# CMOS Widely Tunable Second-Order $G_m$-C Bandpass Filter for Multi-Sine Bioimpedance Analysis

Israel Corbacho ⬤, Juan M. Carrillo *⬤, José L. Ausín ⬤, Miguel Á. Domínguez ⬤, Raquel Pérez-Aloe ⬤ and J. Francisco Duque-Carrillo ⬤

Departamento de Ingeniería Eléctrica, Electrónica y Automática, Universidad de Extremadura, Avenida de Elvas s/n, 06006 Badajoz, Spain
* Correspondence: jmcarcal@unex.es

**Abstract:** A CMOS widely tunable second-order $G_m$-C bandpass filter (BPF), intended to be used in multi-sine bioimpedance applications, is presented. The filter incorporates a tunable transconductor in which the responses of two linearized voltage-to-current converters are subtracted. As a result, the effective transconductance can be continuously adjusted over nearly three decades, which allows a corresponding programmability of the center frequency of the BPF. The circuit was designed and fabricated in 180 nm CMOS technology to operate with a 1.8 V supply, and the experimental characterization was carried out over eight samples of the silicon prototype. The simulated transconductance of the cell can be tuned from 5.3 nA/V up to 19.60 μA/V. The measured range of the experimental transconductance varied, however, between 1.42 μA/V and 20.57 μA/V. Similarly, the center frequency of the BPF, which in the simulations ranged from 500 Hz to 342 kHz, can be programmed in the silicon prototypes from 22.4 kHz to 290 kHz. Monte Carlo and corner simulations were carried out to ascertain the origin of this deviation. Besides, the extensive simulation and experimental characterization of the standalone transconductor and the complete BPF are provided.

**Keywords:** bandpass filter; CMOS; continuous tuning; transconductor; wide programmability





## 1. Introduction

The electrical bioimpedance (BI) technique allows characterizing a biological medium through its electrical properties [1]. A sinusoidal excitation signal is applied to the impedance under test ($Z_{UT}$), and its composition is determined in terms of the magnitude and phase of the signal response. A current is usually preferred as the stimulation signal, to avoid damages in the sample. In particular, currents in the range of μA to a few mA are typically used. In such a case, a voltage has to be acquired across $Z_{UT}$, which is usually achieved by means of an instrumentation amplifier, and subsequently conditioned and processed. The measurement of the biological impedance at a given single frequency is commonly known as bioimpedance analysis (BIA).

Bioimpedance spectroscopy (BIS) is used to determine the impedance over a frequency range, which is known as the dispersion range $\beta$, typically varying from several hundreds of Hz to a few MHz. One approach to address the BIS was to use a stepped-sine excitation current, $i_{EXC}$, and carry out the measurements sequentially while the frequency of $i_{EXC}$ was changed for each analysis. Nevertheless, this solution is only suitable when the physiological phenomenon does not change in time or variations are very slow. In the case of time-variant events, it is recommended to follow a multi-sine bioimpedance analysis, which allows simultaneous bioimpedance measurements at different frequencies. Besides, if the model to adjust the frequency response of the medium under test is known, a very limited number of samples results in being sufficient in most applications [2].

The front-end of a multi-sine BI analyzer, shown in Figure 1, includes the generation and summation of all the excitation signals, from $i_{f_{01}}$ to $i_{f_{0n}}$, which are subsequently applied to $Z_{UT}$. The frequency components in the overall response obtained at $Z_{UT}$ must be

separated, which can be achieved by using a bandpass filter (BPF) with a certain selectivity degree. A suitable technique to implement a monolithic BPF is the $G_m$-$C$ approach [3–10], where the characteristic frequencies of the filter are given by the ratio of a transconductance ($G_m$) and a capacitor ($C$), $G_m$ being an appropriate tuning parameter. Indeed, an electronically programmable transconductor allows a straightforward adjustment of the different characteristics of the BPF, whereas a continuous tuning facilitates the implementation of automatic control loops.

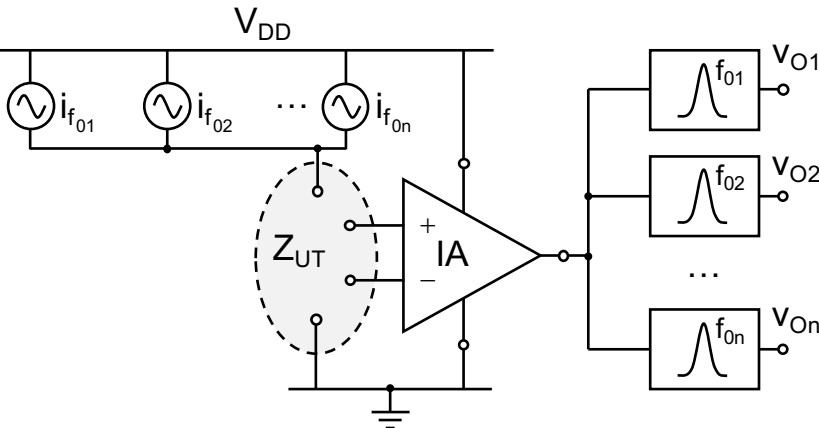

**Figure 1.** Bioimpedance spectroscopy by multi-sine analysis.

In this work, the design of a widely tunable transconductor in 180 nm CMOS technology is presented, and its application to the implementation of a fully integrated second-order $G_m$-$C$ BPF for multi-sine BI analysis is detailed. The proposed transconductor [11] incorporates a mechanism for continuous tuning of $G_m$, which leads to the programmability of the center frequency of the BPF over nearly three decades. Silicon prototypes of the standalone transconductor and the BPF were fabricated, and the correspondence between simulation and experimental results is discussed. The rest of the paper is organized as follows. A review of the state-of-the-art of widely tunable transconductors able to achieve a very low transconductance is reported in Section 2. The principle of operation of the proposed transconductor is described in Section 3, and the structure of the BPF is detailed in Section 4. The simulation and experimental results are reported, compared, and discussed in Section 5, and finally, conclusions are drawn in Section 6.

## 2. Widely Tunable Low-$G_m$ Techniques

The programmability of the center frequency of a BPF over a broad frequency range requires the availability of a widely tunable transconductor. Besides, if operation at low frequencies is required, the minimum achievable transconductance should be very low in order to avoid using excessively large integrated capacitors. Different approaches have been previously proposed in order to design a tunable transconductor able to achieve a very small $G_m$ value [12]. As a result, filters with a broad programmability range [13,14] or suitable for operation at very low frequency have been implemented [15–18].

Multiple solutions have been proposed in the literature to attain a low-$G_m$ transconductor. On the one hand, linearization techniques lead to a reduction of the effective transconductance of a differential stage. Among them, we can mention source degeneration [19,20], bump linearization [21], or triode operation [22]. Furthermore, the current division approach [12] enables increasing the current levels, in order to obtain a good matching, while simultaneously resulting in a very low $G_m$ value. Signal attenuation is another widespread technique used to reduce the effective transconductance of a differential stage. To this end, either passive components [23] or active devices with an inherently attenuating response, such as bulk-driven [24] or floating-gate [25] MOS transistors, can be used. Regarding the tuning of $G_m$, common approaches are based on modifying the biasing current levels of the input section [5] or the drain-to-source voltage,

$V_{DS}$, of the input devices when they are operated in the triode region [26]. Besides, the use of component banks that can be digitally controlled [27,28] or the inclusion of continuously adjustable current mirrors [29] is an extended practice.

In [30], current division and source degeneration techniques were applied to gate-driven, bulk-driven, and floating-gate MOS transistors to obtain a very low $G_m$. Nevertheless, the proposed solutions were not provided with programmability. Mourabit et al. [31] proposed the use of multiple-input floating-gate transistors to implement a transconductor in which linearization was achieved by cancellation of the cubic distortion term. The use of MOS transistors operating in the triode region was proposed in [32,33] to attain a low transconductance value. However, the tuning range can be constrained by the relatively narrow region of the drain-to-source voltage in which a MOS transistor operates in ohmic mode. The channel-length modulation effect can also be exploited to obtain a low transconductance value [34], and this approach has also been adapted to operate in a low-voltage environment [35]. The proposal in [36] combined signal attenuation, current division, and resistive source degeneration to reach low $G_m$ values. In this case, signal attenuation led to an increase of the effective input differential mode (DM) voltage range, which constrained the operation of the common mode (CM) feedback (CMFB) section. Besides, current division requires the use of a certain amount of biasing current, which is not subsequently used for the signal processing. Current steering [18,37] is another solution that allows $G_m$ reduction and electronic tuning in a linearized transconductor. In [38], the bootstrapping effect was applied to a passive resistor, with the help of two voltage followers, to achieve transconductance values in the range of tens of nA/V. Nevertheless, the tuning range obtained was very narrow. Finally, it is worth pointing out that, in addition to the use of bulk-driven [39–41] and floating-gate [3] MOS transistors, also quasi-floating-gate MOS devices [42] have been proposed in the literature to obtain a widely tunable low-$G_m$ transconductor.

## 3. Proposed Widely Tunable Transconductor

The conceptual block diagram of the proposed transconductor is illustrated in Figure 2. As observed, the output current contributions of two matched voltage-to-current ($V$-to-$I$) converters, $G_{m,M}$ and $G_{m,R}$ from the main and replica, respectively, are subtracted by reversing the polarity of their input voltages. Besides, the output current of $G_{m,M}$ is conveyed to the transconductor output by a current mirror with a gain nominally equal to unity, while the current gain for $G_{m,R}$ is equal to a parameter, $\eta$. The resulting effective transconductance is

$$G_{m,eff} = \frac{(1-\eta)i_O}{v_{I,DM}} = (1-\eta)G_m \tag{1}$$

where $G_m = G_{m,M} = G_{m,R}$. If the value of the gain of the current mirror that processes the signal from $G_{m,R}$ is programmed in the range $0 < \eta < 1$, $G_{m,eff}$ can be continuously tuned over a very wide range of values, from $G_m$ down to values very close to zero.

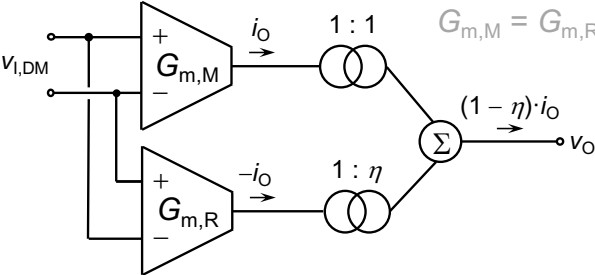

**Figure 2.** Block diagram of the proposed tunable transconductor.

The transistor-level implementation of the proposed transconductor is depicted in Figure 3. The principle of operation of the two *V*-to-*I* cells is based on converting a voltage into a current on a resistor, $R_M$ and $R_R$ at $G_{m,M}$ and $G_{m,R}$, respectively. Two source followers, transistors MI and MS, isolate the resistor in each cell from the preceding and the subsequent stages, while load transistors ML are used to mirror the current signal to the output node. The transconductance of each *V*-to-*I* converter can be expressed as

$$G_m = G_{m,M} = G_{m,R} = \frac{2}{R} \frac{1}{\left(1 + \frac{2}{R} \frac{1}{g_{m,MI}}\right)} \tag{2}$$

where $R = R_M = R_R$ and $g_{m,MI}$ is the transconductance of equally sized transistors MI1 to MI4. The factor of 2 indicates that the current generated is conveyed to the output node through both branches of the *V*-to-*I* converter, whereas the second term describes analytically the loading effect of $R$ on the source followers.

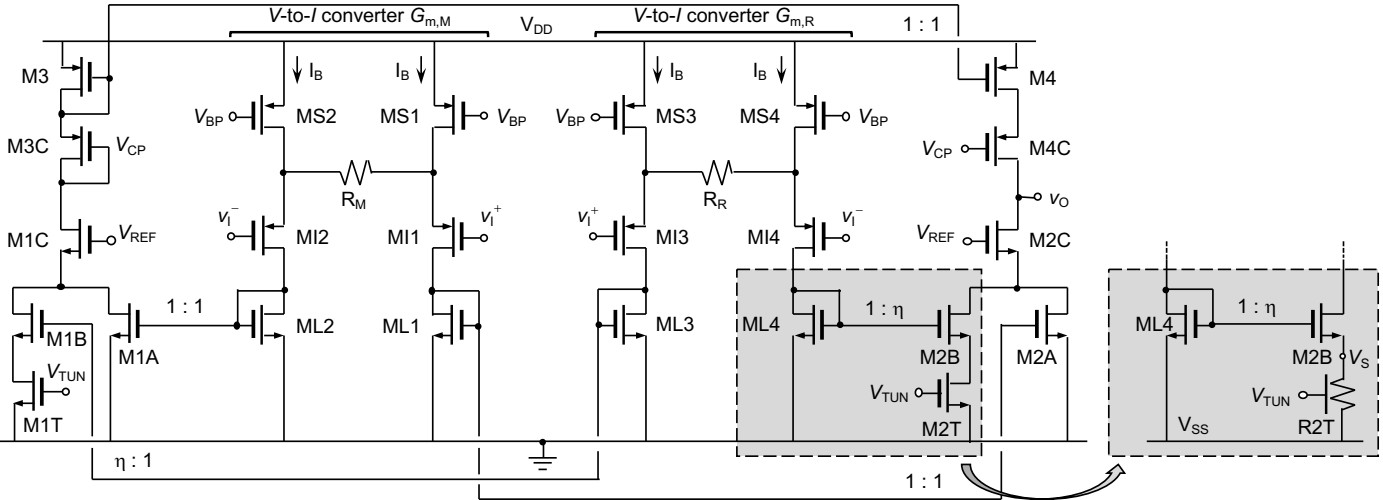

**Figure 3.** Transistor-level implementation of the proposed tunable transconductor.

The input DM voltage range of the transconductor is constrained by the maximum current, $I_B$, that can flow through the source degeneration resistors, that is

$$v_{I,DM}^{max} = \pm I_B \cdot R \tag{3}$$

Besides, the input CM voltage is limited close to ground and $V_{DD}$ by the operation in the saturation of the input transistors, MI1 to MI4, and the current source transistors, MS1 to MS4, respectively. Regarding the output voltage range, it can be extended, if required, by using a fully differential (FD) structure, which is a common practice in low-voltage environments and also provides well-known additional benefits. Besides, in the particular case of the circuit in Figure 3, the use of a FD configuration would allow getting rid of the systematic offset voltage introduced by the PMOS current mirror formed by M3 and M4. Nevertheless, the need for a CMFB network in a differential structure and its design complexity in the case of a tunable circuit led to the choice of a single-ended structure, with which the specifications of the intended application can be met.

The transconductance of the *V*-to-*I* converter $G_{m,R}$ can be continuously tuned by including transistors M1T and M2T below the output transistors of the NMOS current mirrors, M1B and M2B, as observed in Figure 3. Indeed, when the voltage at the gate of transistors M1T and M2T, $V_{TUN}$, is equal to ground, these devices are turned off, and the effective drain-to-source resistance of the transistor is ideally infinite. Hence, only the contribution of $G_{m,M}$ is processed through current mirrors with gain *1:1* in Figure 3. As the value of $V_{TUN}$ increases, the drain-to-source resistance of the tuning devices, represented by the equivalent resistance R2T in Figure 3, is reduced and the voltage at the source terminal

of transistors M1B and M2B, $V_S$, is set closer to ground. For the extreme situation in which $V_{TUN} = V_{DD}$, ideally $V_S \approx 0$ and, hence, the gain of the NMOS current mirrors is very close to unity. In such a case, $G_{m,eff} \approx 0$, which theoretically allows obtaining a very low transconductance value. The gain of the programmable current mirrors, $\eta$, has a nonlinear dependence on the control variable $V_{TUN}$, and its particular expression will depend on the inversion level of the transistors, i.e., weak, moderate, or strong. Nevertheless, the response of the transconductor in terms of total harmonic distortion (THD) is not impacted by this fact as $V_{TUN}$ is first adjusted to obtain the required transconductance level and is then kept constant during the normal operation of the circuit.

It becomes apparent from Figure 2 that a tunable transconductor can be implemented by using only the replica transconductor $G_{m,R}$ and the programmable current mirror with gain *1:η*. Indeed, a DM voltage applied to the input of this arrangement produces an output current equal to $\eta \cdot i_O$, which is equivalent to an effective transconductance $\eta \cdot G_m$, which can be adjusted by means of parameter $\eta$. Nevertheless, the actual implementation of the *V*-to-*I* converter in Figure 3 reveals that, for very low values of $\eta$, the value of the current flowing through the output branch is very small, and hence, the linear operating range of transconductor is significantly reduced as compared to the proposed configuration.

As was already detailed, the principle of operation of the proposed transconductor is based on subtracting the individual transconductance contributions of two ideally equal *V*-to-*I* converters. Therefore, the minimum achievable effective transconductance relies on the matching between these basic building blocks. In principle, there are different possible sources of error, highlighting the key role of the source degeneration resistors and the current mirrors. The sensitivity of the proposed transconductor to random variations in the source degeneration resistors, $R_M$ and $R_R$, was analytically evaluated, and the corresponding results are given next. A similar analysis can be carried out for the mismatch contribution of the current mirrors. Assuming values for the resistors $R_M = R + \Delta R/2$ and $R_R = R - \Delta R/2$, the expression of the minimum effective transconductance, reached for the tuning case in which $V_{TUN} = V_{DD}$, i.e., $\eta = 1$, can be approached as

$$G_{m,eff}^{min} \approx \frac{2}{R} \frac{1}{\left(1 + \frac{2}{R}\frac{1}{g_{m,MI}}\right)} \cdot \left|\frac{\Delta R}{R}\right| \tag{4}$$

That is, the minimum achievable $G_{m,eff}$ moves from the ideal value of zero to approximately a level in the range predicted by (4).

The proposed transconductor has a single-stage structure, and hence, the open-loop voltage gain can be written as

$$A_v(s) \equiv \frac{v_o(s)}{v_{i,dm}(s)} = \frac{(1-\eta)G_m R_{out}}{1 + sC_L R_{out}} \tag{5}$$

where

$$R_{out} = r_{o,M4}\frac{g_{m,M4C}}{r_{o,M4C}} || [r_{o,M2A} || (r_{o,M2B} + r_{o,M2T})] \frac{g_{m,M2C}}{r_{o,M2C}} \tag{6}$$

is the output resistance, $C_L$ is the load capacitor, and only the dominant pole has been taken into account. From (5), the open-loop DC gain, $A_v(0)$, and the dominant pole, $\omega_0$, of the transconductor can be expressed, respectively, as follows:

$$A_v(0) = (1-\eta)G_m R_{out} \tag{7a}$$

$$\omega_0 = \frac{1}{C_L R_{out}} \tag{7b}$$

where $V_{TUN} = 0$ V, the effective transconductance $G_{m,eff}$ takes its maximum value ($G_m$), the DC current flowing through the output branch is equal to $I_B$, and the intrinsic gain of the transconductor at low frequencies displays its maximum value, which is relatively high due to the use of cascode transistors. Nevertheless, for the case in which $V_{TUN} = V_{DD}$, the value

of $G_{m,eff}$ reaches its minimum level, whereas the DC current at the output branch is equal to $2I_B$, that is twice with respect to the other extreme case considered. As a consequence, the open-loop voltage gain of the transconductor will be greatly reduced and its closed-loop response will be correspondingly affected despite the inclusion of cascode transistors. For this reason, a long channel length was set for the transistors determining the open-loop gain of the transconductor, in order to increase its value in the worst-case operating point.

## 4. Second-Order $G_m$-C Bandpass Filter

The block diagram of the second-order $G_m$-C BPF designed and implemented is shown in Figure 4. A routine hand analysis leads to the following expression for its transfer function:

$$H(s)_{BP} = \frac{\frac{G_{M1}}{C_2}s}{s^2 + \frac{G_{M4}}{C_2}s + \frac{G_{M2}G_{M3}}{C_1C_2}} \tag{8}$$

where $G_{Mi}$, with $i$ = 1 to 4, is the effective transconductance of the *i-th* transconductor and $C_1$ and $C_2$ are on-chip capacitors. The gain at the center frequency, $|H(\omega_0)|$, the center frequency, $\omega_0$, and the quality factor, $Q$, can be obtained from (8) and expressed as

$$|H(\omega_0)| = \frac{G_{M1}}{G_{M4}} \tag{9a}$$

$$\omega_0 = \sqrt{\frac{G_{M2}G_{M3}}{C_1C_2}} \tag{9b}$$

$$Q = \sqrt{\frac{C_2}{C_1} \cdot \frac{G_{M2}G_{M3}}{G_{M4}^2}} \tag{9c}$$

It is worth pointing out that the inclusion of four transconductors in the BPF implementation allows programming the voltage gain, the center frequency, and the quality factor independently. The BPF illustrated in Figure 4 aims at the separation of signals at different frequencies in multi-sine bioimpedance analysis, which requires a moderate to relatively high quality factor. The $Q$ of the filter can be robustly determined, in view of (9c), as a ratio of transconductances, i.e., $G_{M2}G_{M3}/G_{M4}^2$, or a ratio of capacitors, that is $C_2/C_1$. In the design described in this work, the second option was selected in order to be able to use the same transconductor to implement $G_{M1}$ to $G_{M4}$. Therefore, considering all the transconductors equal, i.e., $G_{M1} = G_{M2} = G_{M3} = G_{M4} = G_M$, and by setting $C_2 = kC_1 = kC$, the gain at the center frequency, the center frequency, and the quality factor may be obtained from (9) as

$$|H(\omega_0)| = 1 \tag{10a}$$

$$\omega_0 = \frac{G_M}{\sqrt{k}C} \tag{10b}$$

$$Q = \sqrt{k} \tag{10c}$$

The resulting BPF has constant values of $|H(\omega_0)|$ and $Q$, whereas $\omega_0$ can be tuned by modifying the value of $G_M$ in (10b). The programmability of $G_M$ was obtained by adjusting the control variable $V_{TUN}$ in Figures 3 and 4. Indeed, when $V_{TUN}$ = 0 V, the value of $G_M$ reaches its maximum level, $G_M = G_m$, which leads to the highest achievable center frequency. On the other hand, for the case in which $V_{TUN} = V_{DD}$, the value of $G_M$ reaches its minimum value, and hence, also $\omega_0$ in the BPF does. Finally, it is worth noting that the voltage $V_{AGND}$ in Figure 4 represents the analog ground, that is the intermediate level between $V_{DD}$ and ground when a single supply is used.

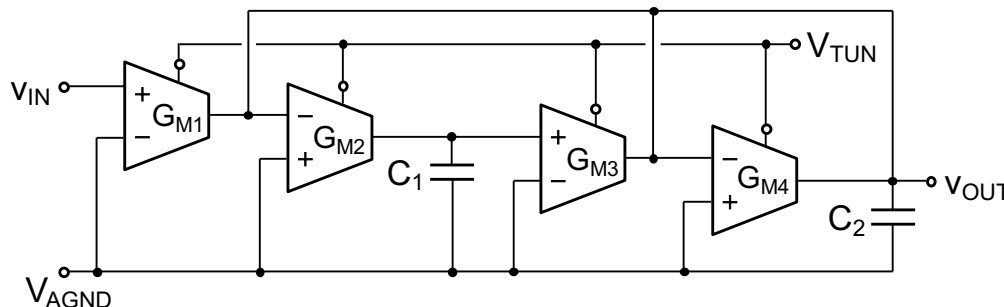

**Figure 4.** Circuit schematic of the designed $G_m$-$C$ BPF.

## 5. Simulation and Experimental Results

The transconductor illustrated in Figure 3 and the second-order $G_m$-$C$ BPF depicted in Figure 4 were designed and fabricated in 180 nm CMOS technology to operate with a single supply voltage of 1.8 V. The transistors' aspect ratios, source degeneration resistors' values, and biasing currents levels were estimated by taking into account the expressions of the transconductance in (1) and (2), so that the filter response can span the intended frequency range and the input DM voltage range in (3), in order to be able to process the typical input signals level, in the order of tenths of mV in our application. Subsequently, these values were refined by means of simulations with a twofold objective: achieving the proposed specifications and ensuring the operation in saturation of all the active devices, thus resulting in the sizes provided in Table 1. A microphotograph of the chip is provided in Figure 5, where the layout of the transconductor and the BPF are detailed. A total of eight samples of the silicon prototypes were experimentally characterized.

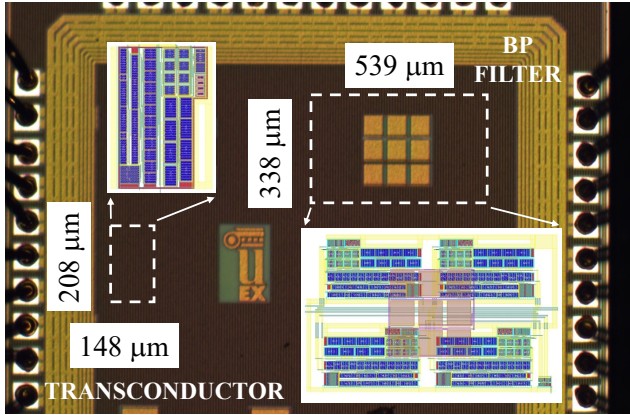

**Figure 5.** Chip microphotograph.

**Table 1.** Transistor aspect ratios of the transconductor in Figure 3.

| Device | W/L (μm/μm) | Device | W/L (μm/ μm) |
| --- | --- | --- | --- |
| MI1-MI4 | 200/1 | MS1-MS4 | 48/1 |
| ML1-ML4 | 150/3 | M1A, M1B, M2A, M2B | 15/3 |
| M1T, M2T | 200/1 | M1C, M2C | 120/3 |
| M3, M4 | 240/3 | M3C, M4C | 240/3 |

In the case of the transconductor in Figure 3, the biasing current, $I_B$, was set to 20 μA and the source degeneration resistors, $R_M$ and $R_R$, were implemented with high-resistivity non-salicided polysilicon with a nominal value of 5 kΩ. The reference voltage $V_{REF}$ used to bias the NMOS cascode transistors was set to 0.9 V, whereas the PMOS cascode transistor M4C was biased by the diode-connected transistor M3C. Besides, it is worth pointing

out that the current mirrors connected to the main and replica *V*-to-*I* converters were sized to have an additional attenuation of *10:1* in order to further reduce the minimum achievable transconductance value. The output terminal of the transconductor was directly connected to the bonding pad, avoiding any internal buffering, with the goal of carrying out a thorough experimental characterization of the standalone block.

The effective $G_m$ of the transconductor was determined by applying an input DM signal superimposed on a CM voltage of 0.9 V, while shorting the output terminal also to 0.9 V, i.e., the mid-supply. First, the output current, $i_O$, was obtained as a function of the input DM voltage, $v_{I,DM}$, as illustrated in Figure 6, where the simulated and experimental responses are provided. A highly linear behavior of the output current can be observed during a substantial range of the input DM voltage. Then, the derivative of $i_O$ with respect to $v_{I,DM}$ led to $G_{m,eff}$, which is represented in Figure 7. A logarithmic scale was selected for the ordinate axis, due to the broad range achieved for the simulated transconductance (Figure 7a), which spans over nearly three decades. Indeed, when $V_{TUN} = 0$ V, $G_{m,R}$ being disabled, a maximum value for the transconductance of 19.92 μA/V was obtained in typical mean conditions (*tt*), whereas for $V_{TUN} = 1.8$ V, $G_{m,R}$ being fully enabled, the transconductance was reduced to only 25.7 nA/V. On the other hand, the measured transconductance (Figure 7b) showed a maximum value of 20.57 μA/V, in close agreement with the simulated result, whereas the minimum value, 1.42 μA/V, was noticeably higher than in the simulations. The $G_m$ measurements were repeated by using a different methodology. A commercial transimpedance amplifier (TIA) was connected to the output of the transconductor, in order to convert the output current into a voltage. The value of $G_{m,eff}$ was determined by deriving the output voltage with respect to the input DM voltage and taking into account the gain of the TIA. The experimental results obtained in this case confirmed the responses illustrated in Figure 7b.

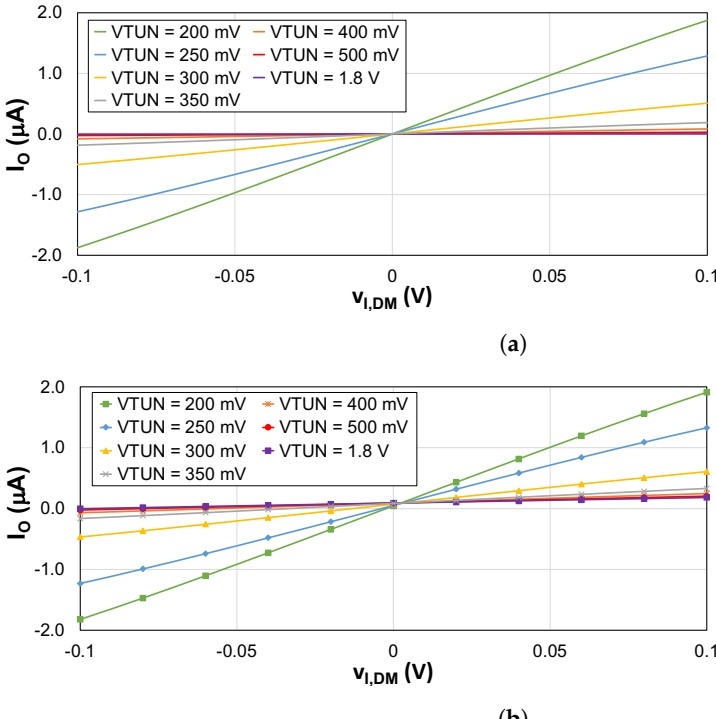

**Figure 6.** Transconductor output current as a function of the input DM voltage: (**a**) simulated and (**b**) measured responses.

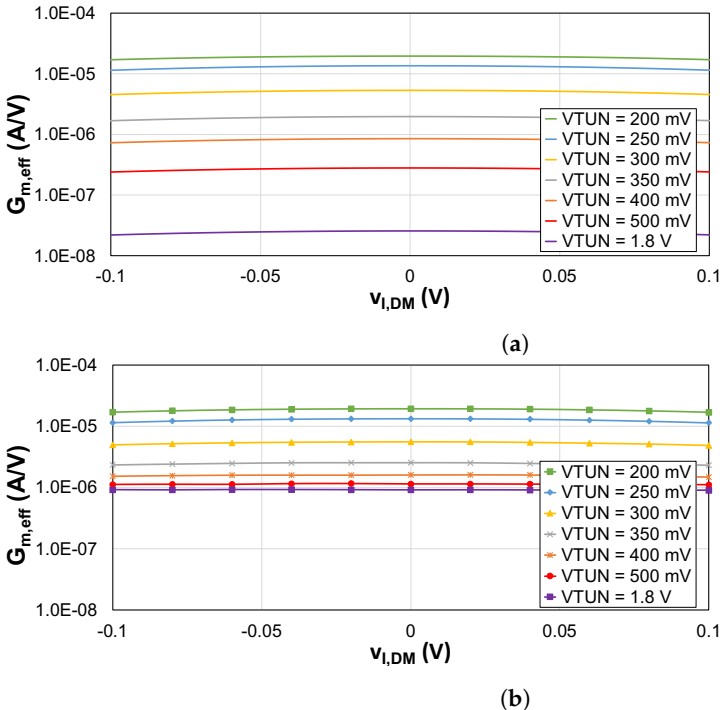

**Figure 7.** Effective transconductance vs. $v_{I,DM}$: (**a**) simulated and (**b**) measured responses.

The impact of mismatches and process variations on $G_{m,eff}$ was assessed. Very similar results were obtained at the corners (*ff*, *ss*, *fnsp*, and *snfp*) as compared to the *tt* case, that is there was not an important change in $G_m$ when the process parameters were equally varied in the same type of transistors. Besides, a 1000-run Monte Carlo analysis, including mismatch and process variations in a 3-$\sigma$ distribution, were carried out for a large variety of values of the control variable $V_{TUN}$. The corresponding results are provided in Figure 8, where the upper error bars, indicating the standard deviation of the effective transconductance, are included for this case. The transconductance simulated in typical mean and post-layout conditions are also included in Figure 8, for comparison purposes and to illustrate the influence of parasitics, respectively. As observed, the experimental results were in general out of the error margin of the Monte Carlo response when the standard deviation was considered to define this range. It is worth pointing out that, even though the statistical simulations showed possible negative values for the transconductance, this fact can be counteracted by the tuning scheme, which grants the possibility of achieving a positive value in every case.

Different sources of error, including systematic and random mismatches, could explain the divergence between the simulated and the experimental response of the effective transconductance, which becomes evident from Figures 7 and 8. Nevertheless, the analysis of the experimental results obtained in the eight available samples of the transconductor showed that, in all the cases, the measured $G_{m,eff}$ presented a maximum level very close to the simulated value and a minimum bound always higher than the value expected from the simulations. This fact suggests that the limitation preventing achieving a very low transconductance could have a systematic origin. Besides, the Monte Carlo analysis reported previously showed that the experimental error of the transconductance was larger than the standard deviation of the simulated results, which was mainly caused by random variations. For these reasons, the inclusion of transistors M1T and M2T in the overall structure, along with a possible underestimation of their on-resistance in the transistors models, was considered as the main factor causing the divergence of the experimental transconductance from the expected response when the transconductor was programmed to generate low $G_m$ values.

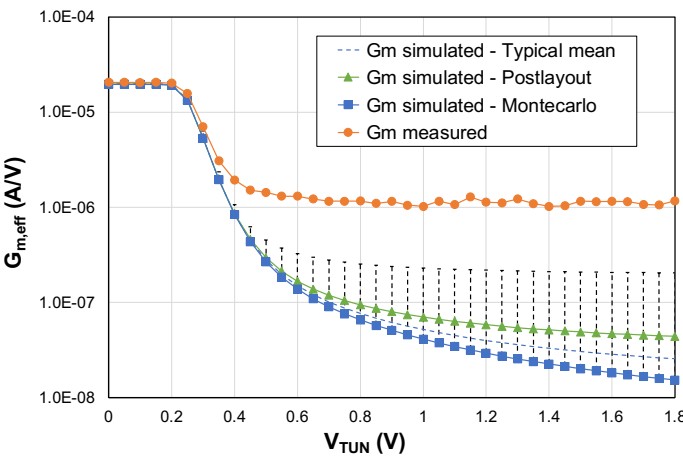

**Figure 8.** Effective transconductance as a function of $V_{TUN}$.

The effect of transistors M1T and M2T on $G_{m,eff}$ is difficult to fully eliminate, even though it can be counteracted by inducing a similar behavior in the current mirrors that are not tuned, that is on those processing the signal of the *V*-to-*I* converter $G_{m,M}$. The scheme proposed in Figure 9 includes a dummy device, M1D, in the current mirrors with a fixed gain in order to reduce their gain in the same proportion as was performed in the programmable current mirrors when $V_{TUN} = V_{DD}$. Indeed, in this particular case, the presence of transistors M1D and M2D ideally canceled out the impact of transistors M1T and M2T on $G_{m,eff}$ when the minimum value of the transconductance was intended to be achieved. As a consequence, the subtraction of $G_{m,M}$ and $G_{m,R}$ was more effective, as it was demonstrated by means of a 1000-run Monte Carlo analysis including 3-$\sigma$ mismatch and process variations, which led to an average value for the effective transconductance of only 1.2 ± 197.4 nA/V. Furthermore, the inclusion of transistors M1D and M2D had a negligible effect when the transconductor was programmed to obtain a higher $G_m$ value. The effectiveness of the solution illustrated in Figure 9 has to be proven experimentally.

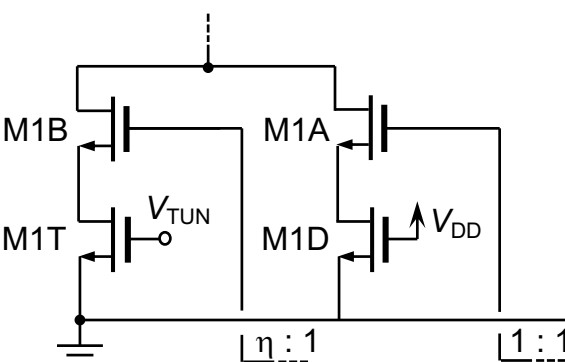

**Figure 9.** Scheme to further reduce the value of $G_{m,eff}$.

The overall performance of the widely tunable transconductor presented is summarized in Table 2 for $V_{TUN} = 0$ V and $V_{TUN} = 1.8$ V. In general, there was good agreement between the simulation and experimental results for high transconductance values ($V_{TUN} = 0$ V). The gap arising when the transconductor was programmed for a low $G_m$ ($V_{TUN} = 1.8$ V) was due to the disparity between the simulated and measured transconductances. The data expressed in Table 2 as the mean value plus/minus the standard deviation were obtained through a 1000-run Monte Carlo analysis with mismatch and process variations in a 3-$\sigma$ distribution in the case of simulations and from the measurements conducted on eight different samples in the case of the experimental results. Besides, the closed-loop configuration was established by shorting the output terminal to the inverting input terminal, that is by connecting the transconductor in a non-inverting unity-gain feed-

back configuration. Additionally, a load capacitor was connected to the output terminal in order to obtain a dominant-pole response and, hence, ensure appropriate frequency and time responses. The total output capacitance was estimated to be 64 pF, including the load capacitor, as well as the capacitive effects of the PCB used for test purposes and the test probe. The relatively high value of the load capacitor reduced the bandwidth of the transconductor, thus leading to a decrease of the noise integrated from the low-frequency range (100 Hz) up to the bandwidth ($\sim$50 kHz). As a result, a high value of the dynamic range (DR) was obtained.

**Table 2.** Comparison of the simulated and experimental performance of the proposed low-$G_m$ widely tunable transconductor.

| Parameter | Simulated | Measured |
|---|---|---|
| $I_{DD}\vert_{V_{TUN}=0\text{ V}}$ ($\mu$A) | 93.1 | 96.3 |
| $I_{DD}\vert_{V_{TUN}=1.8\text{ V}}$ ($\mu$A) | 97.1 | 99.5 |
| $G_{m,eff}\vert_{V_{TUN}=0\text{ V}}$ (A/V) | 19.60 $\mu$ $\pm$ 0.86 $\mu$ | 20.57 $\mu$ $\pm$ 0.30 $\mu$ |
| $G_{m,eff}\vert_{V_{TUN}=1.8\text{ V}}$ (A/V) | 15.3 n $\pm$ 189.4 n | 1.42 $\mu$ $\pm$ 0.40 $\mu$ |
| $V_{os}\vert_{V_{TUN}=0\text{ V}}$ (V) | 3.45 $\mu$ $\pm$ 1.2 m | 1.2 m $\pm$ 3.7 m |
| $V_{os}\vert_{V_{TUN}=1.8\text{ V}}$ (V) | 23.2 m $\pm$ 345.2 m | 52.5 m $\pm$ 41.7 m |
| OL [†] DC gain $\vert_{V_{TUN}=0\text{ V}}$ (V/V) | 3753 | 3802 |
| OL [†] DC gain $\vert_{V_{TUN}=1.8\text{ V}}$ (V/V) | 6.5 | 737 |
| CL [‡] DC gain $\vert_{V_{TUN}=0\text{ V}}$ (V/V) | 0.995 | 0.999 |
| CL [‡] DC gain $\vert_{V_{TUN}=1.8\text{ V}}$ (V/V) | 0.198 | 0.897 |
| BW $\vert_{V_{TUN}=0\text{ V}}$ (kHz) | 48.6 | 51.9 |
| BW $\vert_{V_{TUN}=1.8\text{ V}}$ (kHz) | 0.3 | 3.1 |
| $SR^+/SR^-\vert_{V_{TUN}=0\text{ V}}$ (V/ms) | 88.6/87.9 | 88.9/95.4 |
| $SR^+/SR^-\vert_{V_{TUN}=1.8\text{ V}}$ (V/ms) | 0.039/0.042 | 6.6/3.5 |
| $v_{I,DM}^{max}$ [*] $\vert_{V_{TUN}=0\text{ V}}$ (mV) | $\pm$104 | $\pm$104 |
| $v_{I,DM}^{max}$ [*] $\vert_{V_{TUN}=1.8\text{ V}}$ (mV) | $\pm$90 | $\pm$89 |
| Integrated noise ($\mu$V) | 19.4 | 41.3 |
| DR (dB) | 71.6 | 65.0 |

[†] OL: open-loop. [‡] CL: closed-loop. [*] $f$ = 1 kHz for THD = $-40$ dB.

The performance of the proposed transconductor is compared in Table 3 to other solutions previously reported to simultaneously achieve a low $G_m$ value and a broad programmability range. As observed, the solution presented provided a wide simulated transconductance tuning range at the cost of the highest supply current. Nevertheless, as was pointed out before, this power consumption is a well-known design trade-off to achieve a determined input DM voltage range for a given value of the source degeneration resistor $R$.

The second-order $G_m$-$C$ BPF in Figure 4 was implemented by using four identical replicas of the transconductor shown in Figure 3, $G_{M1}$ to $G_{M4}$, and two integrated metal–insulator–metal capacitors with nominal values of $C_1$ = 2.5 pF and $C_2$ = 20 pF, resulting in a quality factor of $Q = \sqrt{8} \approx 2.83$. The supply voltage used was equal to 1.8 V, and the analog ground voltage was set to mid-supply, i.e., $V_{AGND}$ = 0.9 V, whereas all the transconductors were biased with a current equal to 20 $\mu$A. An on-chip buffer was connected to the output of the filter for testing purposes.

**Table 3.** Performance comparison of the proposed transconductor with other solutions reported in the literature.

| Parameter | [31] TCAS1'05 | [34] MEJ'09 | [36] MEJ'14 | [18] Sensors'19 | [35] Sensors'20 | [38] TCAS2'22 | This Work Simulated | This Work Measured |
|---|---|---|---|---|---|---|---|---|
| Technology (μm) | 0.8 | 0.35 | 0.35 | 0.18 | 0.18 | 0.18 | 0.18 | |
| $V_{DD}$ (V) | 1.5 | ±2.5 | ±2.5 | 1.8 | 1.0 | 1.8 | 1.8 | |
| $I_{DD}$ (μA) | 0.67 | 60 | 32 | 3 | 0.028–0.27 | 2.2 | 93.1 | 96.3 |
| $G_m^{min}$ (nA/V) | 0.46 | 0.03 | 39.5 | 0.5 | 0.62 | 15 | 15.3 | 1420 |
| $G_m^{max}$ (nA/V) | 82 | 2500 | 367.2 | 5000 | 6.28 | 18.5 | 19,600 | 20,570 |
| $G_m^{max}/G_m^{min}$ | 178.3× | 833 k× | 9.3 | 10 k× | 10.1× | 1.2× | 1281.0× | 14.5× |
| Area (mm$^2$) | 0.04 | 0.046 | 0.0061 | 0.014 | 0.0271 | 0.0099 | 0.0308 | |

The DC current consumption of the proposed BPF slightly varied with the value of $V_{TUN}$. Indeed, for levels of the tuning variable close to ground, the signal of the replica *V*-to-*I* converter in each transconductor was not mirrored to the output terminal, thus resulting in a total current of 413.3 μA. When the tuning transistors were completely switched on, for $V_{TUN}$ approximately equal to 0.4 V and higher, the corresponding current mirrors were fully enabled, and hence, the current consumption increased up to 429.1 μA. The total difference between the different operating points was reduced by the use of the *10:1* attenuation factor introduced in order to systematically reduce the value of $G_{m,eff}$. The simulated and experimental DC offset voltages of the BPF were extracted, respectively, from a Monte Carlo analysis (1000 runs with mismatch and process variations) and from measurements on the eight silicon samples of the prototype. The on-chip test buffer used to isolate the filter during the measurements was based on a PMOS source follower, which caused a level shift of the output voltage with respect to the ideal value of 0.9 V (mid-supply). For this reason, only the standard deviation of the output voltage was taken into account, obtaining simulated and measured values of 1.3/428.6 mV and 9.0/83.1 mV, respectively, for $V_{TUN}$ equal to 0/1.8 V. It can be observed that the output voltage deviation was larger for $V_{TUN}$ = 1.8 V, a case in which the transconductance achieved its minimum value and led to the lowest value of the intrinsic gain of the transconductors involved in the implementation. Indeed, the simulated offset voltage showed a larger deviation because the effective transconductance took a significantly lower value.

The frequency response of the magnitude of the $G_m$-*C* BPF is illustrated in Figure 10, measured for different values of the tuning variable $V_{TUN}$. The center frequency varied from 258 kHz down to 22.4 kHz as the tuning variable, $V_{TUN}$, was increased from 0.25 V to 1.8 V. The overdamping observed in the plot corresponding to $V_{TUN}$ = 0.25 V was due to the proximity of the center frequency of the filter to the secondary poles of the transconductors and was boosted by the effect of the value of $Q$ selected. As the level of $V_{TUN}$ was made higher, the filter center frequency was separated from the secondary poles and the overdamping was reduced.

The response of the filter center frequency, $f_0$, was measured over the entire range of the tuning magnitude, $V_{TUN}$, that is from 0 V to 1.8 V, and is depicted in Figure 11. As observed, the center frequency of the BPF behaved exactly equal to $G_{m,eff}$ in Figure 8 when $V_{TUN}$ was swept. Therefore, the BPF center frequency presented similar limitations in the tuning range as the effective transconductance of the proposed transconductor. Indeed, the simulated frequency response of the BPF led to minimum and maximum values of $f_0$ equal to 500 Hz and 342 kHz, respectively, while the experimental characterization of the filter provided an operating frequency range between 22.4 kHz and 290 kHz. The quality factor as a function of $V_{TUN}$ is also represented in Figure 11. Each value of $Q$ was estimated taking into account the measurements of the center frequency and the 3 dB bandwidth of the BPF. For $V_{TUN}$ around ground, the high effective transconductance pushed the

filter center frequency to high values. Thus, the proximity of the secondary poles of the transconductors to $f_0$ resulted in an increase of the effective value of $Q$. As $V_{TUN}$ was increased, the value of $G_{m,eff}$ was reduced, and hence, the influence of the secondary poles was alleviated, thus decreasing the quality factor to a level close to the nominal design value.

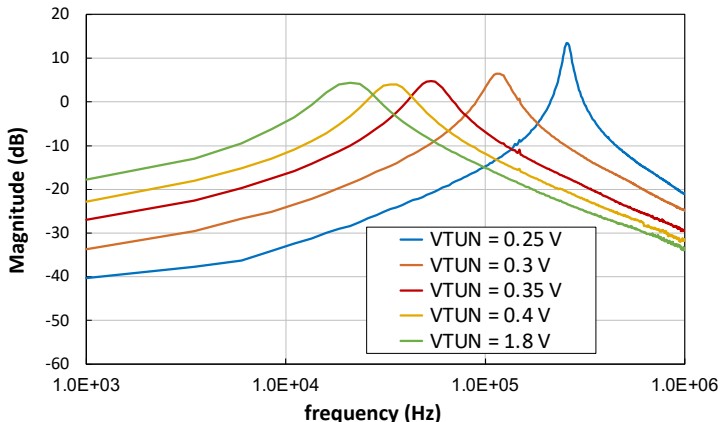

**Figure 10.** Experimental frequency response of the BPF magnitude for different values of $V_{TUN}$.

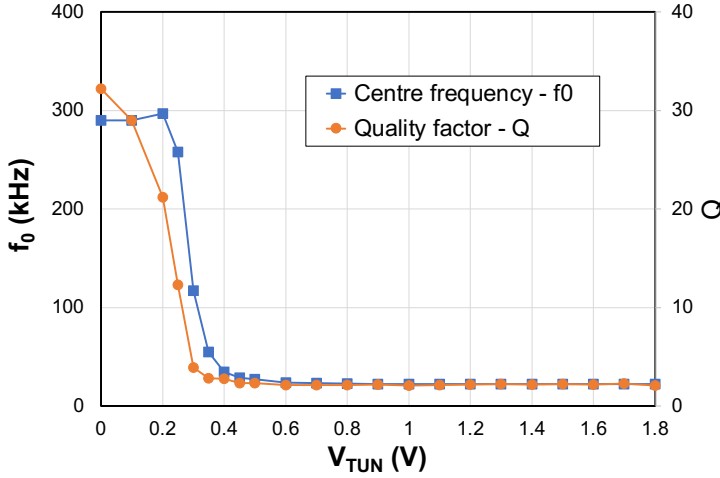

**Figure 11.** Measured values of $f_0$ and $Q$ vs. $V_{TUN}$.

The performance of the proposed second-order $G_m$-C BPF is summarized in Table 4, where it is also compared to other works previously reported. The following figure-of-merit (FoM) [5], which is very widespread, was used to establish an objective comparison:

$$FoM = \frac{P \cdot V_{DD}}{n \cdot f_0 \cdot DR} \tag{11}$$

where $P$ is the power consumption, $V_{DD}$ the supply voltage, $n$ the filter order, $f_0$ the center frequency, and DR the dynamic range. It is worth pointing out two important aspects. On the one hand, a filter implementation of a given order can make use of a different number of transconductors, depending on the degrees of freedom desired to program individually the gain, the center frequency, and the quality factor. Besides, designing a filter to operate at a given frequency allows optimizing the power consumption more efficiently than in the case in which programmability over a broad frequency range is required. For this reason, the FoM in (11) was modified as follows in order to normalize the total power consumption by the number of transconductors incorporated (FoM$_1$) or by the achievable frequency range (FoM$_2$):

$$FoM_1 = \frac{\frac{P}{N_T} \cdot V_{DD}}{n \cdot f_0 \cdot DR} \tag{12}$$

$$FoM_2 = \frac{P \cdot V_{DD}}{n \cdot R_{f_0} \cdot f_0 \cdot DR} \tag{13}$$

where $N_T$ in (12) is the number of transconductors involved in the filter and $R_{f_0} = f_{0,max}/f_{0,min}$ is the ratio of the maximum and minimum achievable center frequencies. As observed in Table 4, even though the proposed solution provided a relatively high value for the initially proposed FoM, it became competitive in terms of the modified comparison parameters, that is $FoM_1$ and $FoM_2$, when the number of transconductors and the achievable frequency range (in the simulated case) were taken into account.

**Table 4.** Experimental performance of the proposed $G_m$-C filter and comparison with similar solutions.

| Parameter | [5] TBCAS'07 | [6] TCAS-II'12 | [7] * MEJ'15 | [8] ICECS'20 | [9] * ICECS'21 | This Work Simulated | This Work Measured |
|---|---|---|---|---|---|---|---|
| Technology (μm) | 0.35 | 0.35 | 0.05 | 0.13 | 0.18 | 0.18 | |
| $V_{DD}$ (V) | 1 | 3.3 | 0.4 | 1.2 | 0.8 | 1.8 | |
| Filter order | 6 | 2 | 2 | 8 | 2 | 2 | |
| $f_0^{min} - f_0^{max}$ (Hz) | ∼100–20 k | 20–20 k | 1–30 k | 2–100 k | 72.7 k | 500–342 k | 22.4–290 k |
| $f_0^{max}/f_0^{min}$ | 200× | 1000× | 30× | 50× | N.A. | 684× | 13× |
| $f_0$ (kHz) | 0.67 | 20 | 10 | 100 | 72.7 | 97.8 | 120.7 |
| Q | N.A | 3 | 1 | 4.8/5.2 | 5 | 3.4 | 3.9 |
| $v_{I,max}$ (m$V_{pp}$) | 40 | 245 ‡ | 178 † | 140 ‡ | 800 | 68 | 64 † |
| In-band noise (μV) | 70.8 | 58.7 | 53.0 | 100 | 266.6 | 196.0 | 212.1 |
| DR (dB) | 49.0 | 63.5 | 68.4 | 49.0 | 60.5 | 41.8 | 40.6 |
| Power (μW) | 44.3 | 75.4 | 31.8 | 256.0 | 24.0 | 713.3 | 767.7 |
| FoM × $10^{-13}$ (SI) | 3.4 | 979.6 | 93.1 | 64.0 | 21.9 | 1571.4 | 1411.6 |
| $FoM_1$ × $10^{-13}$ (SI) | 0.4 | 489.8 | 93.1 | 4.0 | 5.5 | 392.9 | 352.9 |
| $FoM_2$ × $10^{-13}$ (SI) | 0.02 | 0.98 | 3.10 | 1.28 | N.A. | 2.30 | 108.58 |

* Simulated. † @ −40 dB THD. ‡ @ 1-dB compression point.

## 6. Conclusions

Multi-sine bioimpedance analysis requires the use of bandpass filters for signals separation. The use of a BPF with a broad programmability range of the center frequency and able to operate at low frequencies allows using the same filtering section in all the signal conditioning channels. The $G_m$-C approach is a suitable candidate to adjust the characteristics of the BPF in a straightforward manner. A transconductor with widely tunable transconductance and capable of achieving a very low $G_m$ value, thus being suitable to be instantiated in a programmable $G_m$-C BPF, was presented. The principle of operation was based on subtracting the responses of two *V*-to-*I* converters, linearized by means of resistive source degeneration. The proposed transconductor and a second-order $G_m$-C BPF were designed and fabricated in 180 nm CMOS technology to operate with a 1.8 V supply. Measurements from eight samples of the silicon prototype were provided. Deviations between the simulated and experimental results for the ratio $G_{m,max}/G_{m,min}$ of the transconductor (from 1281.0× to 14.5×) and the center frequency range of the filter (from 684× to 13×) were found. Corner and Monte Carlo simulations were used to determine the origin of this disagreement, and a mechanism to further reduce the minimum achievable effective transconductance was proposed.

**Author Contributions:** Conceptualization, I.C., J.M.C., J.L.A. and J.F.D.-C.; methodology, I.C., J.M.C. and J.L.A.; software, I.C. and J.M.C.; formal analysis, J.M.C. and R.P.-A.; investigation, I.C., J.M.C. and J.L.A.; resources, I.C., J.M.C. and M.Á.D.; data curation, I.C., J.M.C., M.Á.D. and R.P.-A.; writing—original draft preparation, I.C., J.M.C., J.L.A. and J.F.D.-C.; writing—review and editing, I.C., J.M.C., J.L.A., M.Á.D., R.P.-A. and J.F.D.-C.; visualization, I.C. and J.M.C.; supervision, J.M.C., J.L.A. and J.F.D.-C.; project administration, J.F.D.-C.; funding acquisition, J.L.A. and J.F.D.-C. All authors have read and agreed to the published version of the manuscript.

**Funding:** This work was supported by MCIN/AEI/10.13039/501100011033 under Project RTI2018-095994-B-I00 and by *Fondo Europeo de Desarrollo Regional (FEDER) "Una manera de hacer Europa"*. Silicon samples granted by EUROPRACTICE MPW and design tool support.

**Data Availability Statement:** Data not subject to confidentiality are available upon request to the corresponding author.

**Conflicts of Interest:** The authors declare no conflict of interests.

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
