# Peer review of "CMOS Widely Tunable Second-Order Gm-C Bandpass Filter for Multi-Sine Bioimpedance Analysis"

_electronics, doi:10.3390/electronics12061326_

Round 1
Reviewer 1 Report
Authors in this research article have presented and investigated CMOS Widely Tunable Second-Order Gm-C Bandpass Filter for Multi-Sine Bioimpedance Analysis. The topic and concept of the paper are interesting and it includes promising results. Prior to final acceptance recommendation the authors are encouraged to address the following comments.
1. Its language needs some minor modifications.
2. Why the authors have compared their article with previous references in two tables
3. It is better to edit the conclusion
4. According to the title of the article (Tunable Bandpass Filter), the author can be discuss more references about the filters. The following references may be helpful.
5. -“Substrate integrated waveguide filter based on the magnetized ferrite with tunable capability”
-“0.6-V 1.65-μW Second-Order Gm-C Bandpass Filter for Multi-Frequency Bioimpedance Analysis Based on a Bootstrapped Bulk-Driven Voltage Buffer”
-“Substrate integrated waveguide filter based on the magnetized ferrite with tunable capability”
Author Response
Thank you for your comments. Please, see the responses in the attached file.

Reviewer 2 Report
The paper is quite well written and the topic is important. Nevertheless I have many remarks regarding sense of the presented amplifier as well as to the presented simulations and measurements. Here are the main notes:
1) The scheme of the tunable transconductor principle is great (Fig. 2), but as you can see, if n factor can vary as stated in the article in the range 0-1, and if we remove the blocks: GmM, 1 to 1 mirror and adder (i.e. 3 blocks out of 5 used), we get the same result and we will use only 40% of the original resources. In addition, we will get less power consumption, so less noise and finally there will be less silicon surface and final cost. Maybe there is a reason for such configuration but I did not found it in the paper.
2) Chapter 3 presents an analysis due to resistors mismatch, while they are not necessarily the main matching problems.
2) Nowhere in the paper are the dimensions of the MOS transistors presented nor the exact production technology used.
3) Simulations as well as measurements of output current in respect to the input voltage io=f(vid) are not presented while they are really important to detect possible problems with linearity as well as with offset.
4) No measurements and simulations of the amplifier/filter offset voltage are shown. No dynamic range or noise for the amplifier is given.
5) The C1 and C2 capacitance values of the filter are not given.
6) Presented transconductance amplifier has single output while for small power supply fully balanced structures are usually employed. Please comment this fact.
In conclusion, taking into account the fabrication of the amplifier and filter, I believe that the article can be published, but before, above mentioned issues must be cleared.
Author Response

(The authors gave the same response as above.)

Round 2
Reviewer 1 Report
The authors have answered the comments of the reviwers.
Reviewer 2 Report
The article has been significantly improved and in its current version its quality is sufficient for publication. Regardless, I draw the authors' attention to:
1) There are many CMOS 180nm fabs, e.g.: AMS, XFAB, TSMC UMC, and many more. Which process was used in the article, despite my comment in the previous review, we did not find out.
2) The supplemented table no. 2 results in very large offsets, which in fact significantly reduce the quality of the presented solution.
3) There are known solutions of CMFB circuits in the literature and their complexity is not large enough in my opinion not to use fully balanced amplifier and filter. The authors did it differently and for research purposes this solution is acceptable.